# What to Do When Accumulated Exposure Affects Health but Only Its Duration Was Measured? A Case of Linear Regression

**DOI:** 10.3390/ijerph16111896

**Published:** 2019-05-29

**Authors:** Igor Burstyn, Francesco Barone-Adesi, Frank de Vocht, Paul Gustafson

**Affiliations:** 1Department of Environmental and Occupational Health, Dornsife School of Public Health, Drexel University, Philadelphia, PA 19104, USA; 2Department of Pharmaceutical Sciences, University of Eastern Piedmont, Novara 28100, Italy; francesco.baroneadesi@uniupo.it; 3Population Health Sciences, Bristol Medical School, University of Bristol, Bristol BS8 2PS, UK; frank.devocht@bristol.ac.uk; 4Department of Statistics, The University of British Columbia, Vancouver, BC, V6T 1Z4, Canada; gustaf@stat.ubc.ca

**Keywords:** measurement error, dose-metric, Bayesian, cumulative exposure

## Abstract

*Background*: We considered a problem of inference in epidemiology when cumulative exposure is the true dose metric for disease, but investigators are only able to measure its duration on each subject. *Methods*: We undertook theoretical analysis of the problem in the context of a continuous response caused by cumulative exposure, when duration and intensity of exposure follow log-normal distributions, such that analysis by linear regression is natural. We present a Bayesian method to adjust duration-only analysis to incorporate partial knowledge about the relationship between duration and intensity of exposure and illustrate this method in the context of association of smoking and lung function. *Results*: We derive equations that (a) describe under what circumstances bias arises when duration of exposure is used as a proxy of cumulative exposure, (b) quantify the degree of such bias and loss of precision, and (c) describe how knowledge about relationship of duration and intensity of exposure can be used to recover an estimate of the effect of cumulative exposure when only duration was observed on every subject. *Conclusions*: Under our assumptions, when duration and intensity of exposure are either independent or positively correlated, we can be more confident in qualitatively interpreting the direction of effects that arise from use of duration of exposure per se. We can use external information on the relationship between duration and intensity of exposure (namely: correlation and variance of intensity), even if intensity of exposure is not available at the individual level, to make reliable inferences about the magnitude of effect of cumulative exposure on the outcome.

## 1. Introduction

We considered a problem of inference in epidemiology when cumulative exposure is the true dose metric for disease, but investigators are only able to measure its duration on each subject. We nest most of our presentation within the context of occupational and environmental epidemiology, while recognizing that the issue also arises in other sub-disciplines of epidemiology. This problem was first highlighted by Johnson who observed that an association with duration can indicate a causal relationship with cumulative exposure when intensity of exposure is independent of its duration, also highlighting that when duration and intensity are inversely associated, a trend with duration can be observed that is in the wrong direction [1]. We are not aware of systematic investigations of correlation structure between duration and intensity of occupational exposures in the context of this problem. However, there is an example of negative correlation between the two, e.g., if new hires are assigned to “dirtier” jobs that then leads them to change employment to avoid such exposure [2]. There are also reports of positive correlations when such feedback is either unlikely [3], or when selection out of the workforce due to high exposures may not be strong [4]. There are settings where duration and intensity of exposure appear to be unrelated within a subject (e.g., for exposures emitted intermittently) [5], and between subjects (e.g., after selection on the basis of vulnerability to exposure, as has been shown to exist in bakers) [6]. Thus, specifics of the workplace, health condition, and selection of the study sample may all influence the correlation of duration and intensity. This raises concerns about both false positive and negative findings that could result from procedures that use duration as proxy for cumulative exposure. De Vocht et al., [4] when intensity and duration had a correlation of 0.3, observed stronger association with cumulative exposure compared to duration alone. Similarly, McDonald et al., [7] reported that cumulative exposure to silica, but not duration alone, was associated with lung cancer, implying that if only duration was available, then the likely causal association would have been missed. Another case in point is the lack of association of cancer mortality with trichloroethylene that may be due to absence of information on exposure intensity [8]. This is suspected, because a finding of an association of trichloroethylene with non-Hodgkin lymphoma was based on cumulative exposure, but was not observed for either duration or intensity alone [9]. Conversely, when an association is reported with duration of exposure and information on intensity is not available, there is a concern that error in exposure due to use of duration as a proxy for cumulative exposure may have created a false positive finding [10,11].

The reason why duration of exposure is sometimes available, but intensity is not, relates to cost associated with assessments of intensity of (workplace) exposure. Duration of exposure is typically derived from employment records or self-reports of occupational histories, which are the minimal requirements in occupational epidemiology. Estimating intensity of exposure requires an additional effort that assigns intensity of exposure to occupational histories and involves estimation processes based on either expert judgments or a typically limited collection of workplace measurements. At best, in most retrospective epidemiological studies researchers have information on the (historic) distribution of exposure intensity, but not individual values. In occupational epidemiology, this led to development of practice and theory of job-exposure matrices [12,13] and group-based exposure assessment [14,15,16]. However, such approaches raise the question of how to proceed with the analysis of health impact of accumulated exposure, when duration is assessed on an individual level (e.g., via questionnaires), while exposure intensity is subject to various modeling assumptions, given that individual-level assessment of exposure intensity is rarely possible (e.g., self-reports of exposure are not reliable, individual exposure measurements are almost always not available). The naive practice in the field has been to compute cumulative exposure indices as if duration and intensity are of equal accuracy, using some form of best guess of intensity, or to resort to analysis by duration of exposure only. The improvement on this practice may lie in framing it in the context of missing data or a measurement error problem.

We considered the problem from the theoretical perspective by exploring the expected behavior of the effect estimates. The focus of our work is not on false positive or false negative occurrences (as would arise from hypothesis-testing) but rather on a more pragmatic path of reasoning in epidemiology that deals with bias and precision of effect estimates as a measure of their usefulness [17,18,19]. For the sake of clarity in describing the key features of the problem, we limit our analysis to the theoretically more tractable situation of continuously measured health outcome suspected be related to the logarithm of cumulative exposure (e.g., relationship of noise to blood pressure [20] or hearing loss [21]), where analysis by linear regression could apply. Such constraints are most directly applicable to cross-sectional studies with continuous exposure and outcome measures (or any design where the time-course of exposure is either not collected, or not relevant to the hypothesis). Thus, we do not address here the problem of time-varying variables. However, working out the details of this relatively simple case is a useful first step towards tackling the problem in more complex study designs, and in other disease models applicable to estimation of effects of exposure on binary and survival-time outcomes. We consider the realistic situation where duration and intensity of exposure may not be independent. Next, using synthetic data motivated by a cross-sectional study of Kennedy et al., [22] we outline and illustrate a Bayesian method aimed at recovering an estimate of cumulative exposure on the outcome, when only duration is assessed for every subject and some information on exposure intensity is available, i.e. is disjointed at the individual (sample) level from duration, following an approach reminiscent of Gustafson and Burstyn [23]. Finally, we illustrate our methodology using data from two waves of the National Health and Nutrition Examination Survey (NHANES) that can be used to assess the association of smoking and lung function. Note that we do not aim to add to the underlying etiological questions, but that this is merely added as a practical example of the proposed methodology.

## 2. Theoretical Analysis of Impact on Estimate of Effect of Cumulative Exposure

For continuously measured health outcome Y_i_ on the i^th^ of n persons, the outcome model is assumed to be:Y_i_ = β_0_ + β_1_*log* C_i_ + e_i_,(1)
where C_i_ is the cumulative exposure, e_i_ is the error term distributed as *N*(0, σ^2^), and σ^2^, β_0_ and β_1_ are the parameters. The cumulative exposure of the i^th^ person is defined as the product of duration of exposure (D_i_) and intensity (I_i_), such that the outcome models can be re-written as: (Y|D, I)~*N*(β_0_+β_1_(*log* D_i_ + log I_i_), σ^2^). There is theoretical and empirical evidence that many occupational exposures are well-described by the lognormal distribution [24,25] and emerging evidence that age up to an event, such as either development of illness or selection into an epidemiologic study, can follow the lognormal distribution [25,26]. Consequently, we focus on situation where (*log* I_i_, *log* D_i_) follows a bivariate normal distribution *N*_2_(**µ**, **Ʃ**), with means µ_I_ and µ_D_, variances σ_I_^2^ and σ_D_^2^, respectively, and a correlation ρ. This assumption is not necessary to linear regression in general, so we are considering a special case where such an assumption is defensible. Mathematical details pertinent to the rest of this section are in Appendix B, while the R [27] code needed to reproduce Figure 1, Figure 2 and Figure 3 is provided in Appendix A.

## 3. Naïve Analysis

The relationships above in equation (1) imply that (Y|D)~*N*(α_0_+α_1_*log*(D), λ^2^), where expressions for (α_0_, α_1_, λ^2^) in terms of the original parameters are given in Appendix B. When the investigators have no information about intensity of exposure and naively regresses outcome on *log*(D) to estimate β_1_ with ${\hat{\alpha }}_{1}$, we show that they incur bias:α_1_ − β_1_ = ρkβ_1_,(2)
where k = σ_I_/σ_D_. (In such an analysis, when the model in Equation (1) is assumed to be true, any interpretation of ${\hat{\alpha }}_{1}$ must be a reflection of the true causal association mediated by non-zero intensity of exposure.) Outside of some uncommon settings (particular combinations of parameter values paired with a very small sample size), this estimator has a root-mean-squared-error (RMSE) greater than that obtained in the complete-data case by the regressing outcome on log(C) exposure to obtain $\hat{\beta }$_1_ (estimate of slope with complete data). In the special case where ρ = 0, bias is not incurred but variance of the estimator is inflated: Var(${\hat{\alpha }}_{1}$) = n^−1^(σ^2^+β_1_^2^σ^2^_I_)/σ^2^_D_ > Var($\hat{\beta }$_1_) = n^−1^σ^2^/(σ^2^_D_ + σ^2^_I_) (general expressions for estimator variances are in Appendix B). This is the same as Berkson-type error when log(D) is used as a surrogate of log(C) with error term log(I)~*N*(µ_I_, σ^2^_I_) [28]. When ρk < −1, the naïve analysis will estimate a target (tend to yield an estimate) that is in the opposite direction from the true effect (Figure 1). In other words, this situation can only occur when (a) intensity and duration are inversely related with sufficiently high correlation and (b) intensity is more variable than duration to a large enough degree to produce ρk < −1, leading to the case highlighted by Johnson [1]. Clearly, in such circumstances, as well as when bias is expected to be substantial, there is a motivation to either collect data on exposure intensity, or use knowledge about the joint distribution of intensity and duration to account for it in data analysis. Furthermore, when the RMSE of a naïve analysis is much worse than that obtainable with cumulative exposure, either further data collection, or adjustment are motivated, such as when duration and intensity are noticeably correlated (e.g., Figure 2). We develop intuition as to whether the adjustment can achieve worthwhile improvements in the next section; it is important to consider this because, where possible, the resources involved in additional statistical analyses and validation studies are less than the cost of full-scale assessment of intensity of exposure.

## 4. Adjusted Analysis: The Limit of What We can Learn when Only D is Available, but ρ and k are Known

We imagine that the investigator can either conduct an exposure measurement campaign, or access existing measurements that yield insights into the relationship between duration and intensity of exposure. This can be done for a subset of subjects, so long as such sample is deemed representative. If we know ρ and k (or more generally know ***µ*** and ***Ʃ***), then it is possible to remove bias but not possible to recover all the precision achievable with complete data. We remove the bias via the relationship implied by equation (2), so the *adjusted* estimator is:(3)$${\hat{\beta }}_{1,A}={\left(1+\rho k\right)}^{-1}{\hat{\alpha }}_{1},$$

We emphasize that this simple form of adjustment arises because the (Y|D) relationship arising from the presumed (Y|I,D) and (I,D) relationships has a simple form. We could arrive at essentially the same adjusted estimator by explicitly casting the problem as a missing-data imputation problem (I must be imputed for all subjects), or as a measurement error problem (D is a surrogate for C with certain properties). That is, the same likelihood function would underpin the inference, whether this is implicit or explicit in the implementation of the estimation scheme. Of course imputation or latent-variable measurement error approaches could still be applied in more elaborate versions of the problem, when a simple form for Y|D is no longer manifested.

The RMSE of the adjusted estimator shows complex behavior relative to the naïve estimator (Figure 3). It must be noted that the adjusted estimator (and its RMSE) are undefined when ρk = −1 (denoted by vertical dotted blue line in Figure 3), and the RMSE tends to very large values near this value (see Appendix B). To develop further intuition about this relationship, we focus on special case of β_1_ = 0 and note that when −2 < ρk < 0, the RMSE of the adjusted estimate is worse than that of the naïve one: although there is no bias, precision deteriorates. This arises when the intensity and duration are inversely related. This is illustrated in Figure 3, that compares RMSE of adjusted and naïve estimators for ρk < 0: the red line indicates where RMSE’s are equal, such that values above the line indicate a situation where adjusted estimators outperform naïve ones. As the strength of the association with cumulative exposure increases (denoted by solid lines in Figure 3, each associated with different β_1_), the range of ρk values that result in worse RMSE in adjusted analysis declines. However, it is noteworthy that the degree to which the naïve estimator can outperform the adjusted estimator is small relative to the advantage of the adjustment under most conditions. The exact shape of solid lines in Figure 3 depends on parameters for which the figure is generated, but Figure 3 depicts the expected general pattern of interdependence of the ratio of RMSE, β_1_, and ρk. Furthermore, the relative magnitude of RMSE grows less favorable for the adjusted estimate for small sample size, because the variance contributes disproportionately to the RMSE, and dwarfs the contribution of bias that plagues the naïve estimator. Conversely, for large sample sizes, variances make little contribution to the RMSE whereas bias remains constant, leading to smaller RMSE for the unbiased adjusted estimator relative to the biased naïve estimator.

The gap predicted by theory between the RMSE values under naïve and complete data analyses that can be narrowed by adjustment tends to be greater when duration and intensity are more strongly correlated (positively or negatively) (Figure 2) and intensity is more varied than duration (large k; not illustrated). In Figure 2, the dotted lines indicate that 95% confidence interval coverage is less than 50%. The confidence interval coverage of naïve analyses degrades with increase in sample size and strength of the correlation between duration and intensity, but tends to be recovered in adjusted analyses. These are the circumstances where we can expect to gain by infusing naïve analyses with knowledge about the joint distribution of intensity and duration. However, when duration and intensity are weakly associated, much more accurate estimates can only be obtained by collecting data on intensity for all subjects (the two middle panels of Figure 2), because the RMSE and coverage of naïve and adjusted data analyses are anticipated not to differ substantially; this also tends to occur when duration is more varied than intensity of exposure (small k; not illustrated).

## 5. Bayesian Analysis when Information of Exposure Duration and Intensity is Disjointed

### 5.1. Models

If some information is available about the distribution of intensity of exposure, then we can learn about the effect of cumulative exposure by combining this with analysis by duration of exposure. In this case, information about duration and intensity is disjointed in the spirit of analysis presented by Gustafson and Burstyn [23] who considered a problem of estimating gene-environment interactions when information on prevalence of exposure was only available at the aggregate level, susceptible genotype was known for all subjects, and it was admissible to assume that susceptible genotype and disease were independent in absence of exposure. In other words, assumptions about the joint distribution of the unobserved quantity (exposure) and the observed quantity (genotype), plus an assumption about the disease model, allowed inference on the joint effect of exposure and genotype. The similarity with the current problem lies in the fact that the measure available on all subjects, i.e., duration of exposure, is associated with the outcome only though the interplay with intensity of exposure, and that information on intensity of exposure is only available in the form of knowledge about the joint distribution with duration of exposure. In other words, in both problems, the use of a mis-specified model allows for the inference about the parameter of interest when specific assumptions are justified.

Let us recall that if we know ρ and k, we can correct for the bias arising from the use of duration as proxy for cumulative exposure and obtain the associated estimator variance, as shown earlier in equation (3). In principle, if we do not know ρ and k but can elucidate informative priors for these parameters, we can sample values from these distributions and incorporate them into Equation (3) to obtain a posterior distribution of β_1_. We use a common default prior for the regression parameters (the g-prior [29], see Hoff [30] for an accessible description). We presume that the investigator uses a scaled beta distribution on [−1, 1] to set the prior on ρ, and a log-normal distribution to set the prior on k. As described in Appendix B, posterior computation is straightforward since the posterior distribution can be shown to be a truncated version of a distribution itself composed of standard distributions. Thus, simple Monte Carlo samples can be drawn from the posterior distribution and Markov chain Monte Carlo methods are not required. The general flavor of this analysis is in keeping with probabilistic bias analysis [19], including the need to discard some samples that violate a constraint imposed on β_1_ by the residual variance of naïve analysis (λ^2^); the proportion of samples that violate the constraint grows as ρk nears −1 (details are in Appendix B).

### 5.2. Synthetic Example

We illustrate this estimation procedure and its properties in synthetic data inspired by a cross-sectional study of the respiratory health of saw-filers by Kennedy et al. [22] In doing so, we simply strive to demonstrate the usefulness of informative priors on ρ and k, not to fully evaluate an existing Bayesian procedure for fitting linear regression. Using linear regression, Kennedy et al. [22] showed a decline in forced expiratory volume in one second (FEV1) in relation to both duration and intensity of exposure (without log-transformation) to cobalt (Co) separately, implying that this association also exists with cumulative exposure. Let us imagine a follow-up study that is about 5 times larger than the original (500 subjects) with similar distributions of duration and intensity of exposure, but without measurements of intensity of exposure to Co due to high cost of obtaining individual measurements. We show how information on the distribution of intensity from the original study can be used to estimate the effect of cumulative exposure in a hypothetical follow-up study. We estimated distributions of duration and intensity from the original paper and set β_0_ and β_1_ to be weaker yet consistent with the original work (see Appendix A for details, including R code for implementation of all analyses). The value of k consistent with the original paper is in the order of 2.6, implying that bias in duration-only analysis can be substantial according to Equation (2). We imagined two plausible values of ρ: −0.5 (e.g., assuming selection of highly exposed workers out of sample available for study due to their deteriorating health) and +0.5 (e.g., assuming a stable workforce with higher exposures in the past); this leads to ρk values of about −1.3 and 1.3, respectively. Both situations are common in occupational and environmental epidemiology and cannot be discounted a priori, but these situations are not meant to be all-encompassing of possible correlations. Having generated synthetic datasets using these parameters, we analyzed them via
the naïve approach (duration only);four wide priors on ρ (two of which admit uncertainty about the sign of the correlation, when the prior mean is one standard deviation below) and k (Priors 1);four narrow priors on ρ and k (Priors 2);assuming known ρ and k; andcomplete data.

The details of implementation in *R* can be found in Appendix A. In both (2) and (3), priors were set such that prior means were either above or below the true values by one prior standard deviation. As such, they represent guesses of various certainty that were off target, as may be expected when priors are reasonably well calibrated, with the best guesses off-target but not so much as to render them blatantly wrong. The results are illustrated in Figure 4 and Figure 5. When ρ = −0.5 and ρk < −1 (Figure 4), we note that the naïve analysis results in a reversal of direction of effect estimate, which is remedied when using the more informative priors, i.e. priors in (3). We observe that 95% credible intervals (CrI) exclude true values in naïve analyses, but capture them in analyses that assume known ρ and k (except in one illustrated case of negative correlation of intensity and duration). When priors are placed on ρ and k, the inference appears to be sensitive to the choice of priors (with inheritance of more uncertainty with broader priors) but is superior to naïve analysis in that it includes the true value in the 95%CrI’s (better coverage). It appears that informative analysis is possible even if there is doubt about the direction of ρ, i.e., priors in (2). Analysis with the narrower priors in (3) tend to yield comparable inference to that obtained with known values of ρ and k. The analysis is clearly challenging when ρ < 0 and k is large, as even knowing these quantities appears to lead to biased inference in some of our synthetic datasets. We repeated all calculations by switching the variances of duration and intensity, leading to k = 1/2.6 = 0.38. As expected, bias in such situation is reduced and the motivation to adjust may be reduced, even where ρ < 0 (Appendix A).

### 5.3. Real-World Application

To illustrate (the advantages of) our methodology, we use the example of a known association between cumulative exposure to cigarette smoke and forced vital capacity (FVC) in the lungs of male adult smokers (currently smoking and restricted to a cumulative consumption of at least 100 cigarettes in life for this example) using the United States NHANES data. Details of data preparation and all calculations (in *R*) are in Appendix A. Information on intensity of smoking (“average number of cigarettes per day during past 30 days”) and duration (“age at survey” − “age started smoking cigarettes regularly”) is available in the 2009–2010 wave of NHANES. We assume that (contrary to the fact) in the subsequent 2010–2012 wave, the decision was made to only collect information on duration of smoking. This would allow us to estimate ρ (= 0.12) and k (= 1.2) from 2009–2010 data (595 persons) and use it to derive priors for analysis of the association between duration of smoking and FVC in 2011–2012 data (570 persons), aimed at inferring the association with cumulative exposure (pack-years). The 2011–2012 data is illustrated in Appendix A. There is evidence of an inverse linear association of log(FVC) with both log(duration) and log(pack-years) of smoking cigarettes, as expected. We note that ρk is equal to 0.14, suggesting that the bias due to use of duration as a surrogate of cumulative exposure is expected to be small. We analyze NHANES data using the same priors (except with different numeric values of ρ and k) as those we employed in the synthetic example with one exception to meaning of a prior previously labeled as “known” is now designated as “fixed” values. To wit, we consider a scenario in which we have the very high confidence that pre-existing data (2009–2010) yielded true values of ρ and k parameters in the 2011–2012 data and use these fixed values for ρ and k. However, it should be noted that even if we have a high confidence of in these values, in this case the values of ρ and k cannot be considered exactly as “known”. The outcome of Bayesian analyses is presented in Figure 6. It appears that in this example the existence of the association and its direction could also be inferred from the use of duration of exposure alone, i.e., there is little gain in terms of the qualitative conclusion by incorporating the additional information on intensity in the 2011–2012 wave. The 95% credible intervals of complete data analysis do not overlap with analyses of incomplete data, even when infused with information on how duration and intensity are related (i.e., ρ and k), except in the case of some wide priors (those among Priors 1). This underscores the challenge of bias-reduction in this specific application, anticipated by theory, due to both small ρ and large value of k (intensity more varied than duration), and argues for importance of quantifying intensity of exposure at individual level. In this application, our method resulted only in a small improvement in the accuracy of the assessment of the strength of the association.

## 6. Discussion

In the context of continuous outcomes amendable to analysis by linear regression, we placed speculations of Johnson [1] about effects of using duration of exposure instead of intensity onto a more solid theoretical foundation and highlighted the importance to bias and precision of the correlation between duration and intensity of exposure, as well as ratio of their variances. Specifically, we stressed the analytical challenges that arise when such correlation is negative, and the intensity is more varied than duration. Lastly, we developed a pragmatic Bayesian approach to the problem.

Our findings are relevant to studies with binary and time-to-event outcomes, although caution is required in drawing analogies. For example, when ρ = 0 and we are reduced to Berkson-type error, logistic regression will be biased towards the null (unlike linear regression) [31] and the situation with Cox proportional hazard model is nuanced with bias depending on rarity of censoring [32,33]. It is perilous to speculate further, given the complexity we discovered in the case of linear regression. We note that the problem we consider falls within larger domain of scholarship on measurement error problem, [34,35] as well as analytical methods for omitted covariates and latent confounders [36,37,38,39], which have advanced solutions for a wider range of models than considered here. It is likely that rapid progress can be made by leveraging such advances where analogy to duration of exposure being a surrogate for cumulative exposure can be defended. At the same time, the mechanics of implementing a Bayesian analysis that we present should be easily adaptable to other study designs and data types, and our approach may inform advances in related statistical problems.

In practice, not only we will be often uncertain about joint distribution of duration and exposure, but also whatever information we have about duration and intensity is typically contaminated by measurement error. This concern is partially addressed when in Bayesian analyses we admit uncertainty about ρ and k, and may discourage analysis that fixes these quantities as “known”. The matter of uncertainty about observed duration of exposure is a more grave concern (e.g., due to missing or inaccurate dates in occupational or residential histories), as it anchors adjustments that are performed via priors on ρ and k. We can try to overcome this problem if there is some information about a measurement error model for duration of exposure, such that duration can be modeled as a latent construct, as in established methods for analyses contaminated by measurement error [34]. However, we note that duration of exposure is usually recorded with reasonable accuracy in occupational epidemiology, at least when employment records are used from traditional industrial environments. Thus, in many circumstances, errors in duration of exposure are likely negligible compared to those in its intensity.

Our findings apply only to situations where the disease model is not mis-specified (e.g., the logarithm of cumulative exposure is the correct dose-metric, there are no lags or thresholds, toxicity is not reversible, the effect is linear in the chosen scale). Where this is not the case, extension of our work to a more flexible modeling approach can be contemplated [40,41], but it is equally important to admit that there is a perpetual uncertainty about the correct dose-metric in epidemiology, even for well-studied problems. As such, any support for a specific dose-metric remains the key element of analysis (e.g., whether the product of intensity and cumulative exposure is the right dose-metric as in [4]) that must precede consideration of duration of exposure as proxy of the true dose-metric [4,42]. Consideration of time-varying measures of duration and cumulative exposure also constitute a natural extension of our work. Where such matters are pivotal, as in analysis of cohort studies, we are willing to speculate that the case of time-varying exposure is not very dissimilar to that which we considered, if viewed from the prism of measurement error problem, in which accumulated exposure up to a given time point or during any discrete time period is approximated by duration of exposure since it start or during a discrete time period.

To circumvent issues involved in the choice of specific functional forms of exposure metrics, such as log(duration) vs. duration per se, many analysts conduct analyses using categories of exposure. Although this is certainly a viable approach, there are concerns associated with such methodology that arise from the induction of differential misclassification of exposure [43,44], increased chance in spurious associations [45] and mis-specifications of disease models when true risks are expected not to have a threshold. Ideally, different functional forms of exposure metrics yield comparable interpretations of the data, with logarithms of duration and cumulative exposure considered because of theoretical properties that we illustrated and because they tend to counteract undue influence of extreme values. 

## 7. Conclusions

When it is reasonable to make assumptions consistent with our work and epidemiologists can be assured that duration and intensity of exposure are either independent or positively correlated, they can be more confident in qualitatively interpreting the direction of effects that arise from the use of duration of exposure in lieu of true dose metrics when the true dose is captured by cumulative exposure. If they can further substantiate a claim that duration of exposure is more variable than its intensity, they can place more weight on inference about the magnitude of true association with cumulative exposure. However, such analyses are unlikely to be found suitable for quantitative risk assessment. To optimize (or in some cases where individual data on intensity is not available, make possible) reliable inference about the magnitude of effects of cumulative exposure on the outcome, epidemiologists can use information on the relationship between duration and intensity of exposure even if intensity of exposure is not available at the individual level.

## Figures and Tables

**Figure 1 ijerph-16-01896-f001:**
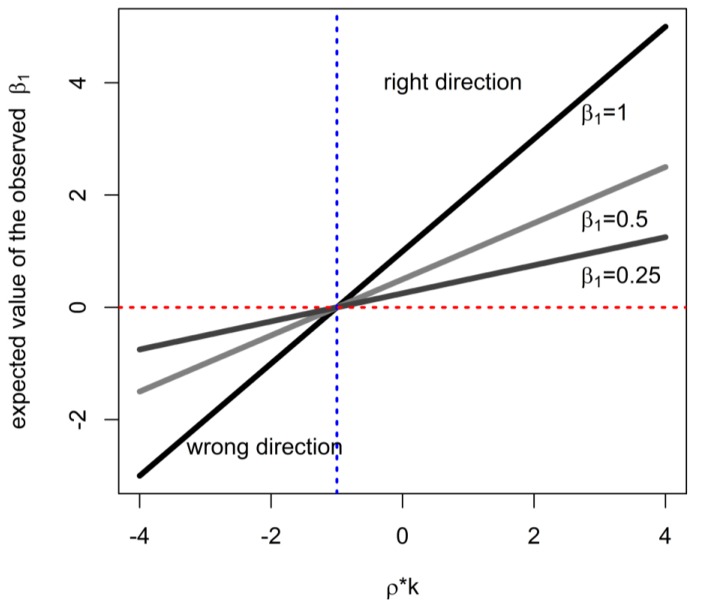
The expected direction of the apparent association with duration of exposure, as a function of correlation of intensity and duration (ρ), ratio of variances of intensity and duration (k), and strength of causal effect (β_1_).

**Figure 2 ijerph-16-01896-f002:**
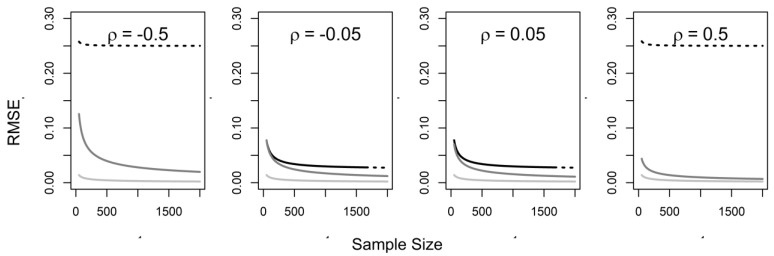
The root mean squared error (RMSE) as function of sample size in analysis (*n*) with duration of exposure (black), duration of exposure adjusted for distribution of intensity (grey), and cumulative exposure (light grey); dotted lines indicate that 95% confidence internal coverage is less than 50%. NB: correlation of intensity and duration varies by panel (ρ), ratio of variances of intensity and duration (k = 1), and strength of causal effect (β_1_ = 0.5).

**Figure 3 ijerph-16-01896-f003:**
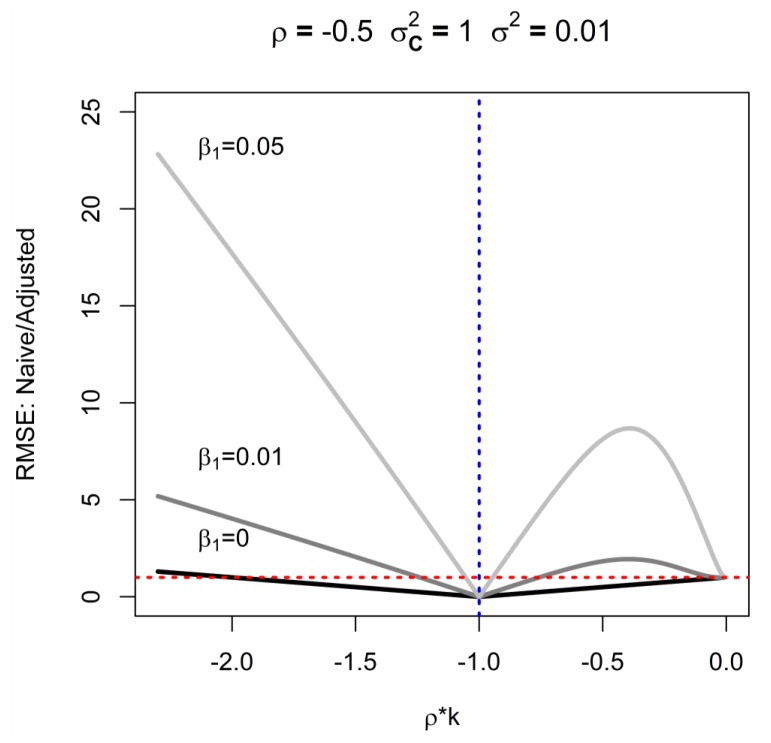
Circumstances when infusion of analysis with additional information on exposure intensity is expected to degrade root mean squared error (RMSE), as a function of correlation of intensity and duration (ρ = −0.5), ratio of variances of intensity and duration (k), and strength of causal effect (β_1_) for *n* = 5000, σ^2^ = 0.01, *Var*(log C) = 1; red line indicates where RMSE’s are equal; blue line indicates where adjusted RMSE is undefined.

**Figure 4 ijerph-16-01896-f004:**
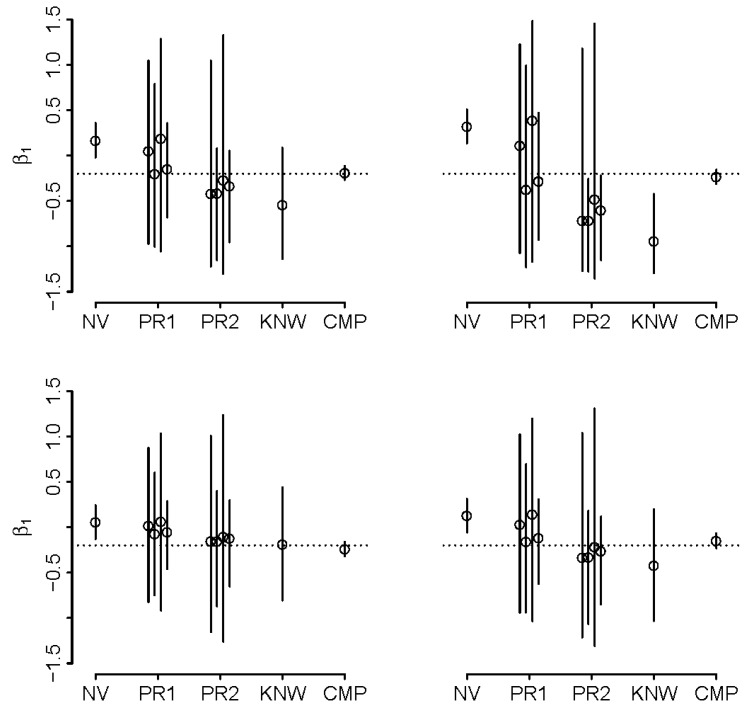
Adjusted estimates of β_1_ with different degrees of knowledge about joint distribution of duration and intensity of exposure when ρ = −0.5 and k = 2.6 in four simulations of synthetic example; naïve estimate (NV) is contrasted with adjusted estimates obtained under “well-calibrated” priors on (ρ,k) that are “wide” (PR1), “narrow” (PR2), estimates obtained with ρ and k known (KNW; the best one can do without complete data), and complete data on intensity and duration (CMP); true value is denoted by dotted line, solid lines represent 95% credible intervals; see text for details.

**Figure 5 ijerph-16-01896-f005:**
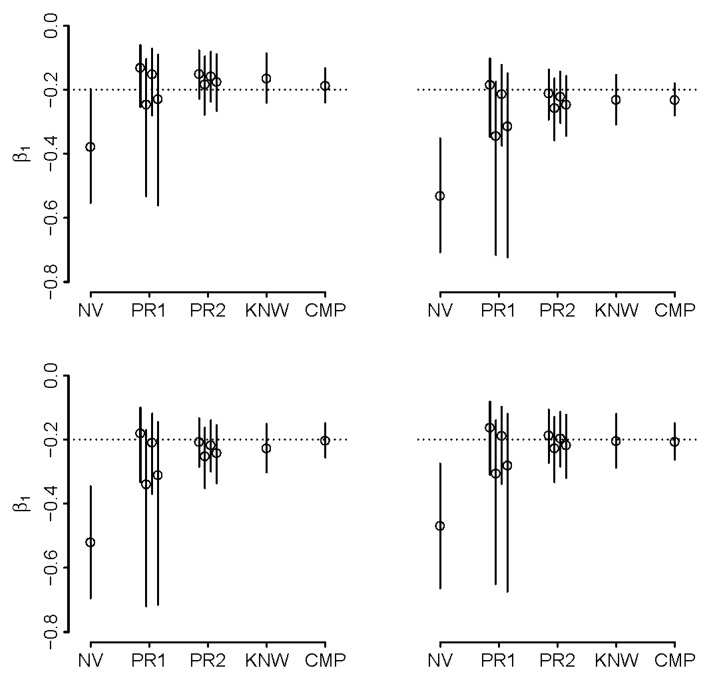
Adjusted estimates of β_1_ with different degrees of knowledge about joint distribution of duration and intensity of exposure when ρ = +0.5 k = 2.6 in four simulations of synthetic example; naïve estimate (NV) is contrasted with adjusted estimates obtained under “well-calibrated” priors on (ρ,k) that are “wide” (PR1), “narrow” (PR2) and estimates obtained with ρ and k known (KNW; the best one can do without complete data), and complete data on intensity and duration (CMP); true value is denoted by dotted line, solid lines represent 95% credible intervals; see text for details.

**Figure 6 ijerph-16-01896-f006:**
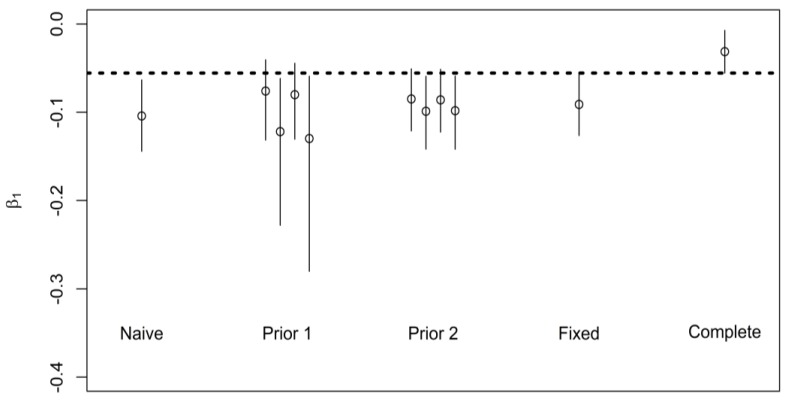
Estimated change in log(FVC, ml) among 570 male current smokers in NHANES 2011–2012 under different priors; naïve analysis is the association with log(years of smoking), complete analysis is the association with log(pack-years), see text for description of different priors (Prior 1, Prior 2, Fixed) that use information on correlation of logarithms of duration and pack-years (ρ) and ratio of standard deviations of logarithms of packs/day and duration (k); circles represent 50th percentile of posterior distributions and line span the 95% credible intervals, dashed line represents lower bound of the 95% credible interval with complete data.

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
