# Peer review of "What to Do When Accumulated Exposure Affects Health but Only Its Duration Was Measured? A Case of Linear Regression"

_ijerph, 2019, doi:10.3390/ijerph16111896_

Round 1

Reviewer 1 Report

General comments:

In this theoretical manuscript, the authors developed a model to investigate the duration and intensity of exposure. Although the rationale for the current paper is essential, however, the manuscript has some minor issues that need to be addressed before it gets to publish. I hope the comments will help the author to make it more clear.

Specific comments:

Abstract:

1)      Results presented in the abstract is not convinced me at all. Can the author present the result in an objective format rather than too subjective?

2)      Conclusion of abstract, grammar issues lines 28.

3)      What type of external information can be used when the intensity of exposure is not available?

Introduction:

4)      Lines 42, Ref 1, is too old, consider it to replace with a recent one.  

5)      If cumulative exposure is multiplied by exposure intensity, how is going to be differently affecting the results? Is that author claiming that these two separate approaches would give us very different results?  If so, how the crude duration of exposure can be calculated for individuals if data/dates are missing?    

6)      Intensity of exposure can be adopted from algorithms. Isn’t’ it? Please explain.

7)      I disagree with the statement from lines 73-76. Please justify your answer.

8)      Lines 228-241, can author explain the models in a simple language rather than making it too complex.   

9)      Lines 252-253, how this can be implemented in cohort studies with continuous follow-ups or within an industry-based analysis of duration of exposure?  

10)  What're new and novel findings here? This should come to the first paragraph in the discussion section.

11)  I can see some limitations, but the author even didn’t mention. This should be in the discussion.

12)  What is the strength of this study?

Author Response

Reviewer 1 (REV1)

REV1: General comments: In this theoretical manuscript, the authors developed a model to investigate the duration and intensity of exposure. Although the rationale for the current paper is essential, however, the manuscript has some minor issues that need to be addressed before it gets to publish. I hope the comments will help the author to make it more clear.

RESPONSE:  We thanks the reviewer for their kind words and have done our best to improve the manuscript using their feedback.

REV1: Abstract: 1)      Results presented in the abstract is not convinced me at all. Can the author present the result in an objective format rather than too subjective?

RESPONSE:  We are not certain what the reviewer found to be “not convincing” and “subjective”, but we reviewed the text and find that it accurately represents our findings.  Please note that we strive to use abstract to draw in the reader rather than to give them essentials of results without encouraging scrutiny of our methods; this is in the spirit of Topical Collection that is considering the article for publication.

REV1: Abstract:  2)      Conclusion of abstract, grammar issues lines 28.

RESPONSE: We revised last sentence starting on line 28; please let us know whether this address the issue alluded to.

REV1: Abstract:  3)      What type of external information can be used when the intensity of exposure is not available?

RESPONSE: This is now captured in the abstract: variance of exposure and its correlation with duration; these can be elicited using any number of ways available in Bayesian methods.  Clarification added.

REV1: Introduction: 4)      Lines 42, Ref 1, is too old, consider it to replace with a recent one. 

RESPONSE: We are not aware of any recent references that define the problem we address [this makes our work even more interesting: we deal with the problem that apparently defied solution for a rather long time] but are prepared to enrich our citations at the suggestion of reviewers.

REV1: Introduction: 5)      If cumulative exposure is multiplied by exposure intensity, how is going to be differently affecting the results? Is that author claiming that these two separate approaches would give us very different results?  If so, how the crude duration of exposure can be calculated for individuals if data/dates are missing?   

RESPONSE:  We do not address the situation where it is reasonable to multiply cumulative exposure by intensity, i.e. D*I^2 exposure metric, but it is dealt with in [4] in different context.  We are not sure that we are able to expand our current work to address a different true disease model (i.e. with D*I^2 as true dose metric) but will mention this explicitly in discussion of correct model specification.  The question of missing dates and errors in duration is of concern in general in epidemiology (as we discuss in the original draft) ; we added specific comment on how this arises when estimate of duration is contaminated by measurement error due to missing or erroneous dates.

REV1:  6)      Intensity of exposure can be adopted from algorithms. Isn’t’ it? Please explain.

RESPONSE:  We are not certain what this comment means.  One can use mean and variance of exposure together with rho, but only information on variance of exposure intensity is required to derive prior on k.

REV1: 7)      I disagree with the statement from lines 73-76. Please justify your answer.

RESPONSE:  We revised text to clarify our arguments and make them more specific to problem at hand.

REV1: 8)      Lines 228-241, can author explain the models in a simple language rather than making it too complex.  

RESPONSE: We already tried to simplify description of Bayesian analysis and placed all the details in Appendix A. We would be great to the reviewer for any specific input she can provide on how to make this paragraph more accessible.

REV1: 9)      Lines 252-253, how this can be implemented in cohort studies with continuous follow-ups or within an industry-based analysis of duration of exposure? 

RESPONSE: When follow-up is continuous, duration of exposure will be related to end of time at risk due to recording of the outcome of interest.  When duration of exposure is based on time worked in an industry or occupation, intensity would have to be specific to that occupation or industry.  These are fundamental features of methods in occupational epidemiology and are not related to our synthetic example that is derived from an occupational cohort with only one job and cross-sectional capture of outcomes.

REV1: 10)   What're new and novel findings here? This should come to the first paragraph in the discussion section.

RESPONSE:  We already say so in the first paragraph of the discussion, staring with (bold text added for emphasis): “In the context of continuous outcomes amendable to analysis by linear regression, we placed speculations of Johnson[1] about effects of using duration of exposure instead of intensity onto a more solid theoretical foundation and highlighted the importance to bias and precision of the correlation between duration and intensity of exposure, as well as ratio of their variances. ..”

REV1: 11)  I can see some limitations, but the author even didn’t mention. This should be in the discussion.

RESPONSE: Paragraphs 2-5 of the discussion all address limitations of our work.

REV1: 12)  What is the strength of this study?

RESPONSE: The strength is in the novel results highlighted in the first paragraph of the discussion and statement of conclusions.

Reviewer 2 Report

REWIEV COMMENTS

The manuscript contains consistent theoretical statistical issues.  The concept of exposure only in term of duration, but not in term of intensity or specific dose-response. It is historically and currently a problem in estimating cumulative exposure by using a simple time regression models. As stated by the authors “When the investigators have no information about intensity of exposure and naively regresses outcome on log(D) to estimate β1 with…..” they incur bias. The design is appropriate for IJEPH and the English is clear and concise the results potentially may be of interest to the readers.  The associated bibliography is generally consistent in this study type and the statistical formalization seems appropriate.
In my opinion, this study has to be accepted for publication with only some remarks.

Small comments for the authors to consider to improve or clarify their study:

### The example used (smoke –lung cancer) need to be specified better, it does not seem very clear. In addition, no similar studies (smoke-lung cancer) are mentioned in the Introduction.  

###The author stated (lines 376-377): “ Our findings apply only to situations where the disease model is not miss-specified (e.g. the logarithm of cumulative exposure is the correct dose-metric, there are no lags or thresholds…..). In a context of dose response there are no measurable effects in terms of lags?

#### SUGGGESTION: the authors have considered a smoke-lung cancer as an example, in this manuscript (see first 21 references) the studies cited refer exclusively to the field of occupational epidemiology. Why not use an example in this field of study?

###The URL (supplementary material) may have been incorrectly typed, or the page may have been moved. Therefore, it was not possible to check the supplementary material, for example the used R-code.

Author Response

Reviewer 2 (REV2)

REV2:  The manuscript contains consistent theoretical statistical issues.  The concept of exposure only in term of duration, but not in term of intensity or specific dose-response. It is historically and currently a problem in estimating cumulative exposure by using a simple time regression models. As stated by the authors “When the investigators have no information about intensity of exposure and naively regresses outcome on log(D) to estimate β1 with…..” they incur bias. The design is appropriate for IJEPH and the English is clear and concise the results potentially may be of interest to the readers.  The associated bibliography is generally consistent in this study type and the statistical formalization seems appropriate.  In my opinion, this study has to be accepted for publication with only some remarks.

RESPONSE: We thank the reviewer for their kind words and recommendation to publish after some revisions.

REV2:  Small comments for the authors to consider to improve or clarify their study:

REV2:  ### The example used (smoke –lung cancer) need to be specified better, it does not seem very clear. In addition, no similar studies (smoke-lung cancer) are mentioned in the Introduction. 

RESPONSE: We did not look at lung cancer in applied example but lung function: that is one of the reasons why lung cancer and smoking are not highlighted in the introduction.  More generally, our work does not depend on any specific exposure-outcome association, so long as our modeling assumptions can be defended.

REV2:  ###The author stated (lines 376-377): “ Our findings apply only to situations where the disease model is not miss-specified (e.g. the logarithm of cumulative exposure is the correct dose-metric, there are no lags or thresholds…..). In a context of dose response there are no measurable effects in terms of lags?

RESPONSE:  We mean that there is no TRUE effect that is lagged and the issue of whether there is power to observe an effect is not relevant in the discussion of model-misspecification.

REV2:  #### SUGGGESTION: the authors have considered a smoke-lung cancer as an example, in this manuscript (see first 21 references) the studies cited refer exclusively to the field of occupational epidemiology. Why not use an example in this field of study?

RESPONSE: We use synthetic example from the field of occupational epidemiology and real world example from environmental epidemiology, thereby stressing wide applicability of our methodological findings.

REV2:  ###The URL (supplementary material) may have been incorrectly typed, or the page may have been moved. Therefore, it was not possible to check the supplementary material, for example the used R-code.

RESPONSE: We supplied all supplemental materials upon submission and regret that these were not available; we ask journal staff to correct the URL (not possible for authors submitting the work) and ask the reviewer to look for supplemental materials submitted with the work.

Reviewer 3 Report

Thank you for the opportunity to review this paper. My comments are as follows:

Abstract: Appropriate, but the conclusion section could be simplified so that the reader can easily see the outcomes of this work.

Introduction: Difficult to follow in parts. Needs a very clear and simplified summary of what the paper is describing. The information is in the text however it is difficult to pinpoint exactly.

Sections 2-5: Appropriately describe the statistics involved.

Discussion: Again, a clear, simple final statement of what this means to the epidemiologist would be helpful.

Author Response

Reviewer 3 (REV3)

REV 3:  Abstract: Appropriate, but the conclusion section could be simplified so that the reader can easily see the outcomes of this work.

RESPONSE: Thank you. We revised the concluding statement to make it easier to understand and are open to other specific suggestions to improve.

REV 3:  Introduction: Difficult to follow in parts. Needs a very clear and simplified summary of what the paper is describing. The information is in the text however it is difficult to pinpoint exactly.

RESPONSE:  We are not sure how to address this.  Opening statements of the introduction tell the reader what we are tackling and the last paragraph of the introduction presents the structure of the paper, as is customary in statistical literature.

REV 3:  Sections 2-5: Appropriately describe the statistics involved.

RESPONSE:  We thank the reviewer for checking our work and confirming that the statistics are appropriately described.

REV 3:  Discussion: Again, a clear, simple final statement of what this means to the epidemiologist would be helpful.

RESPONSE: We do not think that our results can be simplified further but are open to suggestions.  The essence of our findings is in para 1; the rest is devoted to discussion of limitations (these matters are far from simple at best); we segregate conclusions into a distinct section that captures overall message we wish to convey to our peers.

Reviewer 4 Report

Doctor Burstyn I et al. wrote an interesting manuscript about biostatistical advantage and disadvantage of linear regression of using exposure duration as explanatory variable. I would like to present several comments to improve the robustness of the results.

1.      The authors set log-normal model for this research. I would like to request similar simulation in other models (ex. log-logit model, simple linear model).

2.      Figure 3: To know RSME in other settings, I would like to request to other results in changing rho, sigma_C, and sigma.

3.      Discussion of the results of Figures 4-6 might be insufficient. I would like the authors to interpret each result and explain how epidemiologists need to choose the models when they analyse exposure duration.

I think that this manuscript would contribute to the field of epidemiological methodology. I would like to request modifications to improve the study. 

Author Response

Reviewer 4 (REV4)

REV 4:  Doctor Burstyn I et al. wrote an interesting manuscript about biostatistical advantage and disadvantage of linear regression of using exposure duration as explanatory variable. I would like to present several comments to improve the robustness of the results.

RESPONSE: We thank the reviewer for overall positive appraisal of our work.

REV 4:  1.      The authors set log-normal model for this research. I would like to request similar simulation in other models (ex. log-logit model, simple linear model).

RESPONSE: We agree that extension of our work to other disease models, beyond special case of simple linear regression is of interest and plan to pursue such linen of work in the future.  We discuss this matter in second paragraph of the discussion.  Thank you for the encouragement!

REV 4:  2.      Figure 3: To know RSME in other settings, I would like to request to other results in changing rho, sigma_C, and sigma.

RESPONSE: Other settings can be simulated using R code that we supplied. If reviewer is interested in some specific setting, we are prepared to perform relevant calculations and add as supplemental results.

REV 4:  3.      Discussion of the results of Figures 4-6 might be insufficient. I would like the authors to interpret each result and explain how epidemiologists need to choose the models when they analyse exposure duration.

RESPONSE:  Selection of models lies outside of the scope of our work.  We already indicate under what conditions duration is a good proxy of cumulative exposure: positive correlation and small variance of intensity relative to duration.

REV 4:  I think that this manuscript would contribute to the field of epidemiological methodology. I would like to request modifications to improve the study.

RESPONSE: Thank you for your positive reception of our work; we are ready to act on specific suggestions for modification.

Round 2

Reviewer 4 Report

I think that the authors have addressed my concerns. Although there is one concern left, I think that this manuscript would be sufficiently written and discussed for an academic paper. I appreciate the authors for their efforts to report the results of important question in epidemiology.